# How a "Smart regulatory" platform can improve the property fee adjustment dilemma

Dekun Dong, Shuang Li *

Qingdao University of Technology, Qingdao City, Shandong Province, China

* xiaoge2143@163.com

**Data Availability Statement:** All relevant data are within the paper and its Supporting information files.

**Funding:** The author(s) received no specific funding for this work.

## Abstract

In the current process of social and economic development, the cost of living and property management is continuously increasing. In order to maintain the normal operation of property management and provide high-quality services, it is necessary to adjust the property fees reasonably. If the property fees cannot be adjusted for a long time, it will have a detrimental impact on the development of the community. However, the traditional regulatory model has failed to accurately define the quality of property services, leading to information asymmetry and a lack of trust between the two parties, resulting in a deadlock in property fee adjustments. With the rise of intelligent management, a new direction for property fee adjustments has been provided. This article first analyzes the pricing dilemma under traditional regulation through game theory. Then, based on the theory of "Smart regulatory", it proposes the idea of constructing a "Smart regulatory" platform and explores the feasibility of this model using game theory and Matlab simulations. The study found that the "Smart regulatory" platform can resolve information asymmetry and encourage both parties to cooperate in the game. After the establishment of the platform, property fee adjustments are influenced by government supervision, government penalties, and the magnitude of property fee increase.

## I. Introduction

Property management is a market contract behavior [1], in which property service companies provide services and management to homeowners based on contracts, and charge property service fees. Homeowners purchase quality property services by paying service fees, which include maintenance and repairs of houses and facilities, management of common areas, etc., in order to obtain a good living environment and increase the value of their real estate. Although the impact of the property services industry is strengthening, issues such as high operating costs and low efficiency are also emerging [2]. This makes it difficult for property companies to sustain profitable operations in the face of increasingly fierce market competition. For example, in a certain community in Shenzhen, the property fee has been maintained at 2.58 yuan per square meter per month for a long time. However, in 2012 alone, the

**Competing interests:** The authors have declared that no competing interests exist.

community incurred a loss of 3 million yuan [3]. If the property fees cannot be adjusted, the property management company will difficult to maintain the normal operation of their services.

As a regulatory body, the government needs to periodically adjust the guidance prices to meet market demands and maintain the basic operations of property management companies. Additionally, the government should also supervise these companies to ensure they provide high-quality services. However, reality has shown us that traditional regulatory models are no longer able to meet the changing needs of our society [4]. Traditional regulatory models fail to accurately define the quality of property services, leading to information asymmetry and a lack of trust between property companies and homeowners, making it difficult to adjust property fees. Property companies may be forced to lower service quality to cope with rising costs, resulting in a decline in service quality and an increase in conflicts. This difficulty is known as market failure [5], which hinders the ability of property management companies and homeowners to reach mutually beneficial agreements on pricing.

With the continuous development and widespread application of information technology, the digital transformation of the government is becoming increasingly important for the country [6]. Various industries are actively promoting digital transformation to meet the demands and challenges of the digital age. Similar to businesses, the government also faces new opportunities to innovate public services and achieve public governance value through the widespread adoption of digital technology. For example, establishing a digital regulatory platform for government services [7], promoting e-government [8], building smart cities [9], etc. These measures will help enhance government efficiency, optimize public services, improve quality of life, and promote economic development.

This paper based on the theory of "Smart regulation". It analyzes the interests and pricing negotiation dilemma between property management companies and homeowners by constructing a game model and explores the reasons behind the fee adjustment dilemma in property management. Subsequently, "Smart regulatory" is integrated into the game theory model to explore how the government can promote cooperation between the two parties and resolve the pricing dilemma of property fees through the introduction of new technologies. The research demonstrates that leveraging smart regulation helps address deficiencies in government oversight, resolves the challenges of property fee adjustment, and provides guidance and recommendations for other similar pricing dilemma situations.

## II. Literature review

### (1) Relevant studies on government regulation

With the changing times, government regulation is encountering both challenges and opportunities. In the face of ever-changing new environments, government regulation also needs to adapt to technological changes to achieve more precise, systematic, and flexible regulations [10]. Guo and Zhao has reflected on traditional government regulation and believes that the system inertia and functional obstruction caused by the introduction of a hierarchical system within the government have become the main weaknesses in dealing with new social risks [11].

The hierarchical system is an administrative system that classifies and manages government departments based on levels. Under this system, government departments are classified and managed according to hierarchical titles, with horizontal coordination between different levels of government departments and vertical supervision between superior and subordinate departments. However, this hierarchical management structure may lead to efficiency issues in regulatory processes.

Additionally, bureaucratic organizations, in their long-term operation, may selectively and flexibly execute policies based on their own interests, which can also obstruct policy implementation [12].

"Smart regulatory" is an important means of urban governance and innovation in public services. "Smart regulatory" initially emerged from the concept of "Smart governance", which is considered as a holistic and sustainable system of smart development and governance constructed by governments combining technological rationality and governance values [13]. "Smart regulatory" was initially successfully applied in the field of environmental governance, enhancing regulatory efficiency and effectiveness through real-time monitoring and management of regulatory targets [14]. Subsequently, "Smart regulatory" has been applied in various other domains. For example, in the realm of technological innovation and industrial upgrading, "Smart regulatory" can facilitate technology innovation and industrial upgrading by optimizing government funding programs and strengthening collaboration among innovators and innovation labs [15]. In policy-making, the application of "Smart regulatory" theories can provide new solutions. For instance, in the expansion of forests, a pyramid-like approach of multiple policy tools can be gradually upgraded to enhance policy effectiveness and feasibility [16]. In financial supervision, the adoption of "Smart regulatory" theories and blockchain technology can improve the effectiveness and security of securities market supervision [17]. "Smart regulatory" also plays a crucial role in construction supervision and community epidemic prevention, addressing construction management issues, improving efficiency and safety, and enhancing national and governmental risk governance capabilities and systems [18]. In summary, the widespread application of "Smart regulatory" in different domains serves as an important means to enhance regulatory efficiency and effectiveness, and it will continue to play a significant role in the future.

## (2) Property fee-related studies

The adjustment of property service fees aims to ensure the quality of property services and the survival and development of property service companies, while also meeting the demands of homeowners for improved service quality.

Currently, our country is still in a stage of rapid social and economic development, accompanied by an overall increase in wages and prices. For example, the average monetary wage index for urban employed personnel in 2020 compared to 2010 is 266.69%, and the urban retail price index for building materials and hardware and electrical materials in 2020 compared to 2010 is 111.44%. As buildings last longer, the cost of routine maintenance increases. As a labor-intensive industry, property service companies need to periodically and reasonably adjust the prices of property services to ensure profitability and maintain service quality without reducing it. If the adjustment needs of property service companies cannot be met, they may face the risk of long-term operating losses, leading to the eventual withdrawal of companies. This could result in unmanaged communities and a state of chaos [19].

In addition, Research has found that in situations where property service fees are low, property companies often deliberately and selectively ignore negative feedback from homeowners. This practice exacerbates conflicts between homeowners and property companies [20]. For example, in 2021, a district in Qingdao received nearly 7,000 complaints. Among the 237 communities under the jurisdiction of this district, 113 communities had property fees lower than 0.6 yuan, accounting for 47%. Therefore, long-term low property fees are not conducive to community governance, and property fees need to be adjusted.

However, there are currently some difficulties in adjusting property fees. On the one hand, the pricing of property services usually follows government-guided market bidding quotations

and is determined by contracts, making it impossible to independently adjust prices [21]. On the other hand, there is currently a lack of scientific accounting methods that can comprehensively reflect the value of property services [22], resulting in a lack of persuasive property service fee pricing and difficulty in gaining the trust of homeowners [23]. In addition, homeowners lack the right to know about property service fees and are unclear about the income and expenditure of property service fees. These problems result in information asymmetry, making it difficult to adjust property service fees [24].

### (3) Game theory

Game theory belongs to the scope of economics. theory, and its core concept is "Game". It investigates the decision-making behavior and outcomes of decision-makers in interactions. The basic assumptions in game theory include the assumption that all individuals are rational and that all participants have common knowledge of the event. However, with the development of game theory, people have started to question the "Rationality assumption" in game theory and have begun to focus on the "Irrational behavior" in games.

Irrational behavior is a key field in behavioral economics, and early research on irrationality was primarily derived from philosophy. The theory of irrational economics was possibly first proposed by Akerlof and Dickens (1982). They applied irrational models to standard market behavior and advocated that the existence of cognitive dissonance in the market helps explain why citizens resist the government but support government policies [25]. Subsequently, Xia explored the value and essence of irrational behavior [26], while economist Fehr examined the impact of financial shocks on price adjustments through experimental observations. Discovered that individuals exhibit bounded rationality and are influenced by various motivations and factors in economic decision-making [27]. This further introduced the concept of irrational behavior into the field of economics. Kahneman, Slovic, and Tversky proposed three typical forms of irrational behavior: representativeness bias, availability bias, and anchoring and adjustment bias [28]. Hirshleifer observed the behavior of securities market regulators and identified several irrational factors, including the notable effect, neglect of bias, scapegoating, reciprocity, overconfidence, emotional influence, and utility cascades [29]. These factors further guide the direction of research on irrational behavior.

In terms of game theory, researchers such as Camerer C and Wolpert D [30,31] have studied the irrational factors in games and proposed that individual decision-making in games is bounded rationality. They have also conducted research on bounded rationality in games. Basu K analyzes the traveler's dilemma problem and proposes that irrational factors in the game often defy the regular patterns of rational game strategy choices. Instead, they choose strategies that can enhance overall benefits, thereby promoting cooperation with other participants in the game and increasing the benefits for all [32]. In general, irrational factors in game theory are different from what people typically understand as emotions. They encompass a broader range that includes the emotional concepts that influence people's decision-making. Therefore, considering irrational factors in games can make research more realistic and applicable to real-life situations.

Review of existing literature indicates that traditional regulatory models suffer from systemic inertia and functional barriers, which easily lead to information asymmetry. This problem is the main reasons for the difficulties in adjusting housing prices. Moreover, prolonged low fee models in communities are detrimental to community governance. Government needs to take measures to address the issue of information asymmetry. Smart regulation is a new regulatory model that can solve the problem of information asymmetry, especially in solving social problems, and provides opportunities to solve the dilemma of property fee adjustments.

The application of smart regulation provides a good opportunity to break through the dilemma of property fee adjustments. However, there is currently no complete theory in the literature that studies how to solve the dilemma of property fee adjustments from an economic perspective.Currently, applying game theory to study the dilemma of property fee adjustments may be a feasible approach. Game theory falls under the realm of economics. By utilizing game theory, we can analyze decisions and behaviors among participants in situations with information asymmetry. It can assist the government and other stakeholders in better understanding each other's interests and motivations, and finding a balanced solution. Based on this, this paper uses game theory to analyze the causes of the dilemma of property fee adjustments and the operation of smart regulation on property fee adjustments, and provides references and suggestions for solving the dilemma of property fee adjustments.

## III. The cooperation game model between homeowners and property management under traditional supervision

This chapter constructs a game-theoretic model of cooperation between the two parties under traditional regulation and subsequently analyzes the reasons for the dilemma of property fee adjustments.

### (1) Basic model assumptions

Assumption 1: There are two types of game participants in the $\Omega$ game system of homeowners and property: homeowner $A$ and property manager $B$. Both game participants are risk-neutral and not fully rational.

Assumption 2: In the cooperation game system $\Omega$, the "Cooperation" and "Betrayal" decisions of the property manager are to adopt the "Quality service" strategy and the "Cost-saving" strategy, respectively. Their main interests are the property fees charged and the benefits of cost-saving. Homeowners also have two strategies: "Agree to price adjustments" and "Disagree to price adjustments".

Assumption 3: When homeowners choose the "Cooperation" strategy, they need to pay a property fee of $C_1$, and the property company's income is $Q_1$. When homeowners choose the "Betrayal" strategy, they need to pay a property fee of $C_2$, and the property company's income is $Q_2$. When the property company chooses the "Betrayal" strategy, the cost saved by adopting the "Cost-saving" strategy is $D$. When both parties choose "Cooperation", it brings additional income of $A$ and $B$ to the homeowner and the property company respectively. The payoff matrix for both parties is shown in Table 1.

### (2) Game analysis

The property management process requires profitability, so there is a desire to adjust property fees, while homeowners also hope to receive better services. Therefore, both parties will choose to play a game based on their own interests during the contract period. Assuming the homeowner's discount factor is $\delta_1$ and the property management company's discount

**Table 1. The bilateral game of traditional government regulation.**

|  |  | Property management | |
|---|---|---|---|
|  |  | Cooperation | Betrayal |
| Homeowners | Cooperation | $(A - C_1, Q_1 + B)$ | $(-C_1, Q_1 + D)$ |
|  | Betrayal | $(-C_2, Q_2)$ | $(-C_2, Q_2 + D)$ |

factor is $\delta_2$ (The larger $\delta$ is, the more patient the participant is), a subgame perfect Nash equilibrium of (Cooperate, Cooperate) can only be achieved if the following two conditions are met:

$$\sum_{i=1}^{\infty} (Q_1 + B)\delta_1^{i-1} \geq Q_1 + D + (Q_2 + D)\delta_1^1 + (Q_2 + D)\delta_1^2 + ..... \qquad (1)$$

$$\sum_{i=1}^{\infty} (A - C_1)\delta_1^{i-1} \geq -C_1 + (C_2)\delta_1^1 + (C_2)\delta_1^2 + ..... \qquad (2)$$

After simplifying:

$$\delta_2 \geq \frac{D - B}{Q_1 - Q_2} \qquad (3)$$

$$\delta_1 \leq \frac{A}{C_1 - C_2} \qquad (4)$$

Let $(Q_1 - Q_2) = \Delta Q, (C_1 - C_2) = \Delta C$

$$\delta_2 \geq \frac{D - B}{\Delta Q} \qquad (5)$$

$$\delta_1 \leq \frac{A}{\Delta C} \qquad (6)$$

Analysis for the above equations:

1. The homeowner's patience is influenced by $A$, $\Delta C$, where the larger $A$ (The cooperative benefits of both parties), the larger the right side of the formula, and the easier it is for both parties to "Cooperate"; and the smaller $\Delta C$ (The increase in property fees), the larger the right side of the formula, and the easier it is for both parties to "Cooperate".

2. The patience of the property management company is influenced by $D$, $\Delta Q$, $B$, where the smaller $D$ (the benefit obtained from cost savings), the smaller the right side of the formula, and the easier it is for both parties to "Cooperate"; and the larger $\Delta Q$ (the profit obtained after adjusting property fees), the smaller the right side of the formula, and the easier it is for both parties to "Cooperate".

3. In reality, both homeowners and property management companies are rational and seek to obtain maximum economic benefits. Homeowners prefer smaller increases in property fees, but the property management company may increase fees to increase profit. The property management company hopes to gain more profits from adjusting property fees, but the homeowner wants to pay less, which can lead to a smaller increase in property fees that the homeowner can accept and reduce the property management company's profits. The benefits of cooperation between both parties are not always apparent, and the cooperative game between both parties can easily evolve into a "prisoner's dilemma". in which both parties "Betray" each other.

## IV. Evolution of the cooperation game model between homeowners and property management under "Smart regulatory"

### (1) Theoretical analysis

Based on the previous analysis, it is evident that there is a serious problem of information asymmetry between homeowners and property management under traditional regulation. Homeowners often cannot ensure that high-quality services will be maintained after the adjustment of property fees, and it is also difficult for them to have full trust in the property management. The deficiencies of traditional regulation may result in property management companies reducing service quality in pursuit of additional profits, or even resorting to improper means. In this situation, both parties tend to accept decisions that benefit themselves the most, leading to a "prisoner's dilemma" where property fees are difficult to adjust. Based on this analysis, this paper proposes that the government should establish a "smart regulation" platform to address the dilemma of price adjustments. The specific ideas are as follows:

The government plays a crucial role in community governance and should be responsible for establishing a publicly available information disclosure platform to ensure transparency in property service quality and costs [33]. At the same time, property companies should regularly report their detailed income breakdowns and the deployment status of community staff to this platform. The government can assess the quality of property services based on this data and provide reasonable market-adjusted prices. Homeowners can use the platform to report property issues to the government, which can conveniently collect and organize these complaints, conduct regular inspections based on homeowner feedback, and impose penalties on underperforming property companies. The specific operational concept of the platform in this paper is shown in Fig 1.

### (2) Construction of game model

Assumption 4: The government uses the theory of "Smart regulatory" and relies on a big data platform to establish an information service platform. A game evolution model is constructed after that. It is assumed that at a certain point in time, both the property owner and the property management are playing a game regarding the adjustment of property fees, and both parties have bounded rationality.

Assumption 5: In the game process, the property management's decisions are "Betrayal" (Poor service) and "Cooperation" (Quality service), and the property management's main

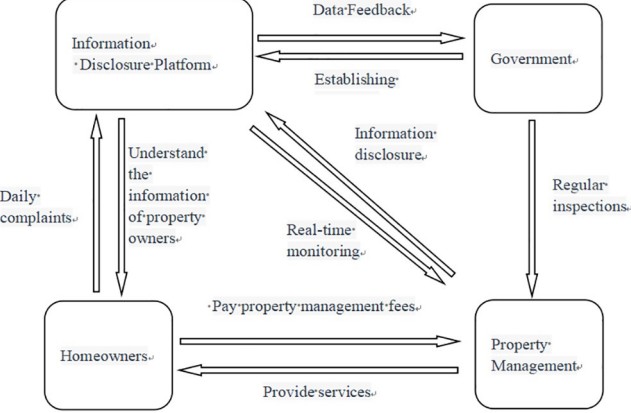

**Fig 1. Government-led platform establishment for regulatory.**

interests are the property fees collected and the abnormal income obtained due to cost savings. Similarly, the property owners also have two strategies: "Cooperation" (Agree to price adjustment) and "Betrayal" (Disagree to price adjustment). It is assumed that the probability of the property management adopting the "Cooperation" (Quality service) strategy is $Y$, and the probability of the "Betrayal" (Poor service) strategy is $-Y$; the probability of the property owners choosing the "Cooperation" (Agree to price adjustment) strategy is $X$, and the probability of the "Betrayal" (Disagree to price adjustment) strategy is $1-X$.

Assumption 6: After introducing the information fair sharing platform, both parties have symmetric information. Property owners can see the quality of the property's services from the data. When the property adopts the "Cooperation" strategy, the property owner's income is $R_1$, and when the property adopts the "Betrayal" strategy, the property owner's income is $R$. On the property management side, due to the transparency of its own service quality, there is a risk of being monitored by the government when adopting the "Betrayal" strategy, and it will be punished by the government with a penalty $H_2$ if detected. At the same time, if the property improves its service quality in the negotiation game, but the property owners choose to "Betray" when the property proposes a reasonable price adjustment, the property owner group will also suffer losses such as reputation, with a penalty $H_1$. The payoff matrix for both parties is shown in Table 2.

The variables represent the following meanings: $C_1$ is the property fee when homeowners choose "Cooperation", $C_2$ is the property fee when homeowners choose "Betrayal" ($C_1 > C_2$); $Q_1$ is the revenue of property management when homeowners choose "Cooperation", $Q_2$ is the revenue of property management when homeowners choose "Betrayal" ($Q_1 > Q_2$); $R_1$ is the quality of service provided by property management when they choose "Cooperation", $R_2$ is the quality of service provided by property management when they choose "Betrayal" ($R_1 > R_2$); $D$ is the abnormal income obtained by property management when they choose "Betrayal", $A$ is the additional income brought to homeowners when both sides choose "Cooperation", $B$ the additional income brought to property management when both sides choose "Cooperation", $H_1$ is the reputational damage suffered by homeowners when they choose "Betrayal", while $H_2$ is the punishment suffered by property management when they choose "Betrayal". Based on these variables, we can construct the payoff matrix for the current game between homeowners and property management.

Expected and average benefits of Homeowners decisions:

$$U_{Y1} = Y(R_1\text{-}C_1 + A) + (1\text{-}Y)(R_2\text{-}C_1) = R_2\text{-}C_1 + Y(R_1\text{-}R_2 + A) \tag{7}$$

$$U_{X1} = Y(R_1\text{-}C_1\text{-}ZH_1) + (1\text{-}Y)(R_2\text{-}C_2\text{-}ZH_1) = R_2\text{-}C_2\text{-}ZH_1 + Y(R_1\text{-}R_2) \tag{8}$$

$$\overline{U_1} = X(U_Y) + (1\text{-}X)(U_X) = R_2\text{-}C_2 + Y(R_1\text{-}R_2) + X(C_2\text{-}C_1 + YA + ZH_1) \tag{9}$$

**Table 2. Evolutionary game between homeowners and property management after the platform is built.**

| | | Property management | |
| --- | --- | --- | --- |
| | | Cooperation | Betrayal |
| Homeowners | Cooperation | $R_1$–$C_1$+$A$, $Q_1$+$B$ | $R_2$–$C_1$, $Q_1$+$D$–$ZH_2$ |
| | Betrayal | $R_1$–$C_2$–$ZH_1$, $Q_2$ | $R_2$–$C_2$–$ZH_1$, $Q_2$+$D$–$ZH_2$ |

Expected and average benefits of property company decisions:

$$U_Y = X(Q_1 + B) + (1\text{-}X)Q_2 = Q_2 + X(Q_1\text{-}Q_2 + B) \tag{10}$$

$$U_X = X(Q_1 + D\text{-}ZH_2) + (1\text{-}X)(Q_2 + D\text{-}ZH_2) = Q_2 + D + X(Q_1\text{-}Q_2\text{-}ZH_2) \tag{11}$$

$$\overline{U} = Y(U_Y) + (1\text{-}Y)(U_X) = Q_2 + D + X(Q_1\text{-}Q_2\text{-}ZH_2) + Y[\text{-}D + X(B + ZH_2)] \tag{12}$$

Combination formula for both sides:

$$F(X) = DY/DT = X(1\text{-}X)(C_2\text{-}C_1 + YB + ZH_1) \tag{13}$$

$$F(Y) = DX/DT = Y(1\text{-}Y)(\text{-}D + XB + ZH_2) \tag{14}$$

## (3) Analysis of strategy evolution stability

To examine the evolutionarily stable strategy of homeowners, let $\frac{dx}{dt} = 0$ Two equilibrium points are: $X_1 = 0; X_2 = 1; Y = \frac{(C_1 - C_2 - ZH_1)}{A}$

Proposition 1:

1. when $Y = \frac{(C_1 - C_2 - ZH_1)}{A}$, $F(X) = 0$, all Y are stable states of the system (9).

2. When $Y > \frac{(C_1 - C_2 - ZH_1)}{A}$, $F'(0) > 0$, $F'(1) < 0$, so $X_2 = 1$ is the evolutionary steady of system (9).

3. When $Y < \frac{(C_2 - C_1 - ZH_1)}{A}$, $F'(0) < 0$, $F'(1) > 0$, so $X_1 = 0$ is the evolutionary steady of system (9).

Proposition 2:

1. When $X = \frac{(D - ZH_2)}{B}$, $F(Y) = 0$, all X are evolutionary stable states of the system (11).

2. When $X > \frac{(D - ZH_2)}{B}$, $F'(0) > 0$, $F'(1) < 0$, so $Y_2 = 1$ is the evolutionary steady state of the system(11).

3. when $x < \frac{(D - ZH_2)}{B}$, $F'(0) < 0$, $F'(1) > 0$, so $Y_1 = 0$ is the evolutionary steady state of system (11)

According to the analysis proposed by Friedman [34], the stability of the equilibrium point of the game can be determined by the local stability of the Jacobi matrix.

$$J = \begin{bmatrix} (1 - 2X)(C_2\text{-}C_1 + YA + ZH_1) & X(1\text{-}X)A \\ Y(1\text{-}Y)B & (1\text{-}2Y)(\text{-}D + XB + ZH_2) \end{bmatrix} \tag{15}$$

According to the assumptions, the points obtained after the evolution need to fall within $V = \{(x, y)|0 \leq x \leq 1, 0 \leq y \leq 1\}$ to be meaningful, so $C_1\text{-}C_2 < B$ and $\text{-}D < ZH_1\text{-}B$ must be satisfied. Let the determinant of the matrix be det(J) and the trajectory of the matrix be tr(J). There exist five equilibria in the conditional range:

$$A = (0, 1), B = (0, 0), C = (1, 1), D = (1, 0), E = (X_0, Y_0)$$

Where $X_0 = \frac{(C_2 - C_1 - ZH_1)}{A}$, $Y_0 = \frac{(D - ZH_2)}{B}$.

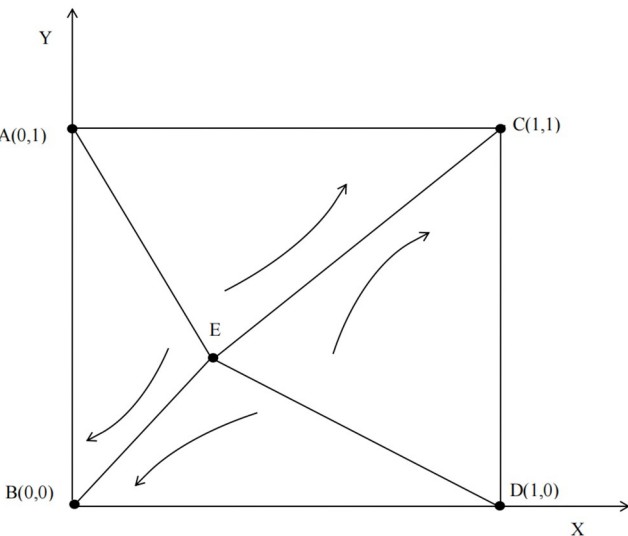

**Fig 2. Phase diagram of the game between homeowners and property management.**

According to the results of evolutionary stability, B(0, 0) and c = (1, 1) are two of the evolutionary stable strategy (ESS) equilibrium points, indicating that both sides choose to "Cooperate" or "Betray"; A = (0, 1) and D = (1,,0) two instability points indicating that both sides will make different decisions; E = (X0, $Y_0$) is a saddle point. The evolutionary phase diagram of property-owner cooperation is shown in Fig 2.

The points $B(0, 0)$ and $C(1, 1)$ are two stable outcomes. When the evolution of the game converges to $B(0, 0)$, the game eventually evolves to "Betrayal". When the evolution of the game converges to point $C(1, 1)$, the game eventually evolves to "Cooperation". The point $E = (X_0, Y_0)$ is the key to the direction of the game.

If the initial state of the two players in the game is near point $E$, slight changes will change the dynamic evolution result of the two players in the game. The final direction of the game participants depends on the area $S_1$ of region $ABDE$ and the area $S_2$ of region $ACDE$. When $S_2 > S_1$, the game is more likely to evolve towards "Cooperation", and vice versa, when $S_2 < S_1$, the game is more likely to evolve towards "Betrayal". The stability of the strategy of both players choosing "Cooperation" can be analyzed by calculating the size of the area $S_2$.

$$S_2 = 1 - \frac{1}{2}\{[(C_2 - C_1) - \frac{ZH_1}{A}] + [\frac{(D - ZH_2)}{B}]\} \tag{16}$$

It can be seen from the formula of $S_2$ that the parameters affecting the area size are $Z$, $C_2$–$C_1$, $H_1$, $H_2$, $D$, and $B$. By taking partial derivatives of these parameters, using "+" to indicate a positive correlation and "−" to indicate a negative correlation, the results is shown in Table 3.

Among them, $D$ and $C_1$-$C_2$) are negatively correlated. When the property size is larger or the rate of increase in property fees is higher, the $S_2$ area is smaller, and the probability of

**Table 3. Analysis of the parameter effects on the choice of cooperative strategy for game participants.**

| Parameters | Partial derivatives | Effect on $S_2$ |
|---|---|---|
| $Z, A, B, H_1, H_2$ | >0 | + |
| $D, (C_1 - C_2)$ | <0 | - |

homeowners choosing the "Cooperative" strategy is smaller, making it more difficult for both parties to "Cooperate". $H_1$, $H_2$, $B$ and $Z$ are positively correlated, that is, the greater the government's punishment and supervision, and the greater the additional benefits after "Both parties cooperate", the larger the $S_2$ area, and the easier it is for both parties to "Cooperate"

## V. Numerical simulation of evolutionary game models

Game theory analysis can help us understand pricing dilemmas and factors that influence strategy choices. Simulation analysis of the game can further enhance our understanding of strategy choices and the impact of government intervention on property service fee adjustments. In this chapter, we utilized Matlab2018a to simulate the equilibrium evolution trajectory of game players with different initial values after the establishment of the information platform, predicted the changes in decision-making before and after the establishment of the platform. We also examined the effects of government supervision, Government Punishment and property fee increase on game strategy choices and evolutionary paths. Through simulation analysis, we were able to identify optimal strategies in the pricing dilemma and evaluate the effectiveness of government intervention, providing valuable insights for further research and policy development.

### (1) Initial value analysis

In evolutionary game theory, the initial values of the game affect the outcomes of the behavioral subjects and the convergence points of the evolutionary process. It is particularly important to find the saddle point $E$ that affects the convergence of both parties' evolution. Based on actual research conducted in a community in Qingdao, The Initial Parameters is show in Table 4

The impact of initial strategy values on the evolutionary results is shown in Fig 3, and the saddle point $E$ is roughly located at (0.2, 0.7). This indicates that homeowners are more willing to improve the community environment and are therefore more likely to accept "Cooperation", but the quality of property services cannot be guaranteed. In contrast, property companies have an information advantage and it is difficult for them to choose "Cooperation". The government does need to establish a regulatory mechanism and adopt a punishment mechanism for regulation, which not only protects the rights of homeowners, but also has a supervisory effect on property companies, thereby solving the "Price adjustment dilemma".

### (2) The role of information platform establishment in the evolution of the game

Before delving into the hypothesis, let's examine the role of the information platform. If the platform fails to produce the intended results during the simulation process, the subsequent discussions would be ineffective. Hence, as a preliminary step, we conducted a simulation analysis to compare the game's evolutionary trajectories between homeowners and property companies under two scenarios: with and without the establishment of the information platform. In the absence of a platform, quantifying property information becomes challenging, and the government faces obstacles in fulfilling regulatory functions. We defined the following

Table 4. Initial parameters.

| Initial conditions | Z | B | $H_1$ | $H_2$ | D | A | ΔC |
|---|---|---|---|---|---|---|---|
| Value | 0.25 | 1.2 | 4.5 | 3.7 | 1.2 | 1.5 | 2 |

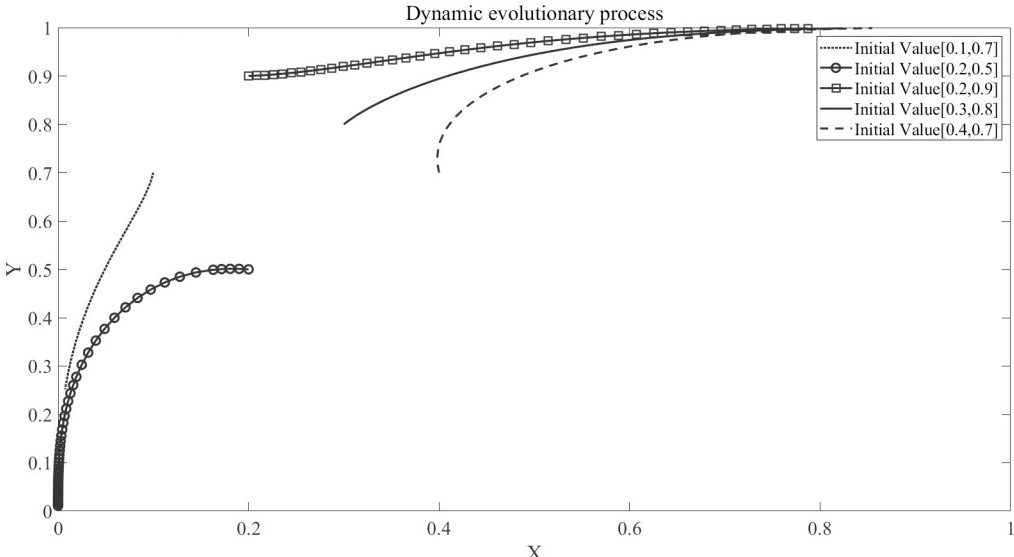

**Fig 3. The impact of initial strategy values on the evolutionary results of the game.**

parameters for our simulation:

$$Z = 0, B = 1.2, H_1 = 0, H_2 = 0, D = 1.2, A = 1.5, \Delta C = 2, (0.5, 0.5)$$
$$Z = 0.25, B = 1.2, H_1 = 4.5, H_2 = 3.7, D = 1.2, A = 1.5, \Delta C = 2, (0.5, 0.5)$$

Fig 4 shows the simulation results of the evolutionary game between the two parties under the scenarios of government platform establishment and non-establishment. The results indicate that in the absence of an information platform, the evolutionary stable strategy for both homeowners and real estate companies tends towards $(0, 0)$, and the system's evolutionary

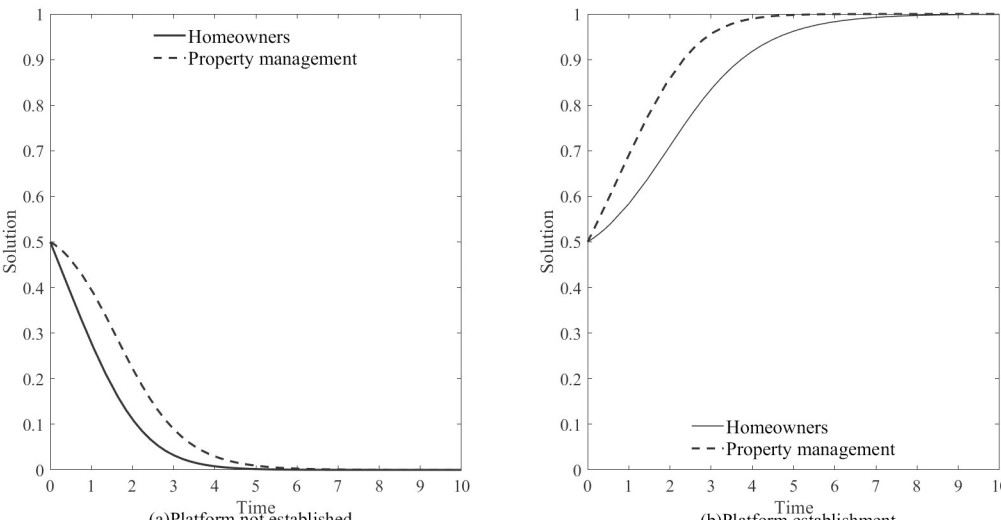

**Fig 4. Game Evolution Before and After Platform Construction.**

stable strategy is "Betrayal". However, when the information platform is established, the evolutionary stable strategy for both parties converges to (1, 1), and the system's evolutionary stable strategy is "Cooperation". The establishment of the information platform enhances the government's regulatory capacity, forcing players adopting betrayal strategies to bear certain punishment risks. Therefore, for their own interests, they choose "Cooperation". The establishment of the information platform plays an important role in resolving the "Price adjustment dilemma" and improving the government's regulatory capacity.

## (3) The impact of various factors on the game between homeowners and property companies

The previous analysis indicates that the government regulatory platform plays a significant role in overcoming the pricing dilemma of property fees. In order to further explore how the government can promote property fee adjustments through the regulatory platform, it is necessary to further discuss the influencing factors in the game. As shown in Table 3, the government's regulatory cycle, penalty intensity, and the magnitude of property fee increases all have an impact on the game between the two parties. Therefore, it is necessary to conduct a detailed discussion and analysis of these factors.

1. The impact of regulatory cycle on the game between the two parties.

According to Table 3, the evolutionary game is influenced by the government's regulatory strength, denoted as $Z$. Therefore, we will discuss the evolutionary process of the game between the two parties under the states of "Weak regulation" and "Strong regulation" by the government. The impact of regulatory strength on the game is defined as follows:

$$Z = 0.25, B = 1.2, H_1 = 4.5, H_2 = 3.7, D = 1.2, \Delta C = 2, A = 1.5, (0.5, 0.5)$$
$$Z = 0.16, B = 1.2, H_1 = 4.5, H_2 = 3.7, D = 1.2, \Delta C = 2, A = 1.5, (0.5, 0.5)$$

Fig 5 show the evolution of the game under government regulatory strength, when the government is in a state of "Weak regulation", it takes a longer time for both parties to reach cooperation because the property owners cannot timely understand the property situation due to

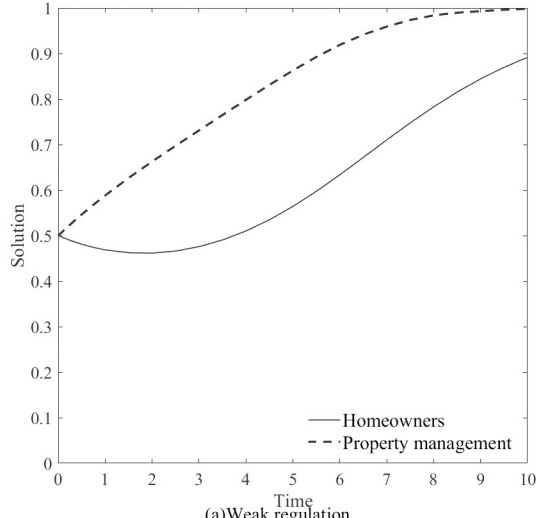
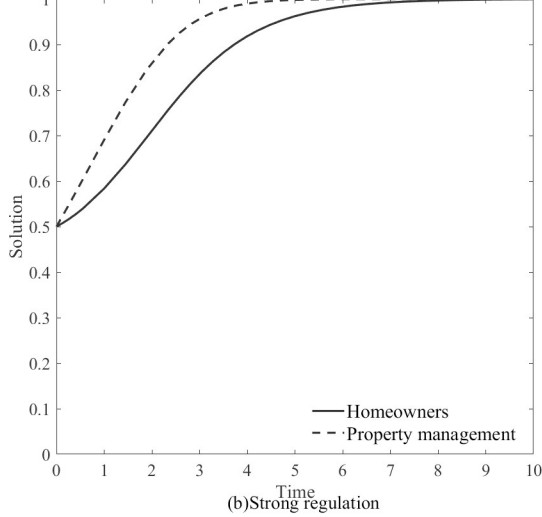

**Fig 5. Evolution of the game under government regulatory strength.**

weak regulation, resulting in hesitancy in decision-making. On the other hand, when the government is in a state of "Strong regulation", the probability of properties choosing "Betrayal" and being discovered by the government increases. In order to protect their own interests, the properties choose "Cooperation". With the assurance from the government, property owners are also more willing to choose "Cooperation", leading to faster achievement of cooperation. Therefore, the government should appropriately strengthen regulatory measures to facilitate cooperation between both parties.

2. The impact of Government Punishment on the game between the two parties.

According to Table 3, the evolutionary game is impact by the government's punishment intensity. As a result, we will examine the evolutionary process of the game under the modes of "Weak punishment", "Conventional punishment", and "Strong punishment" respectively. The impact of Government punishment on the game is defined as follows:

$$\Delta C = 2, Z = 0.25, B = 12, H_1 = 3.6, H_2 = 3, D = 1.2, A = 1.5, (0.5, 0.5);$$
$$\Delta C = 2, Z = 0.25, B = 12, H_1 = 4.5, H_2 = 3.7, D = 1.2, A = 1.5, (0.5, 0.5);$$
$$\Delta C = 2, Z = 0.25, B = 12, H_1 = 5.1, H_2 = 4.1, D = 1.2, A = 1.5, (0.5, 0.5)$$

Fig 6 show the evolution of the game under government punishment, when the government utilizes the platform for regulation, the evolutionary stable strategy for both parties gradually tends towards (1, 1), indicating a choice of "Cooperation"for both parties. In the case of the "Low punishment" mode, the stable strategy is (0, 0), indicating a choice of "Defection" for both parties. When the government adopts a "Strong punishment" mode, the stable strategy is (1, 1), indicating a choice of "Cooperation". Under the strong punishment mode, both parties are able to achieve "Cooperation" more quickly. In summary, the government's use of the platform for regulation promotes "Cooperation" between both parties, and stronger punishment measures expedite the achievement of "Cooperation".

3. The impact of property fee increase on the game between the two parties.

According to Table 3, the evolutionary game is influenced by the property fee increase. As a result, we will examine the evolutionary process of the game under the modes of "Low

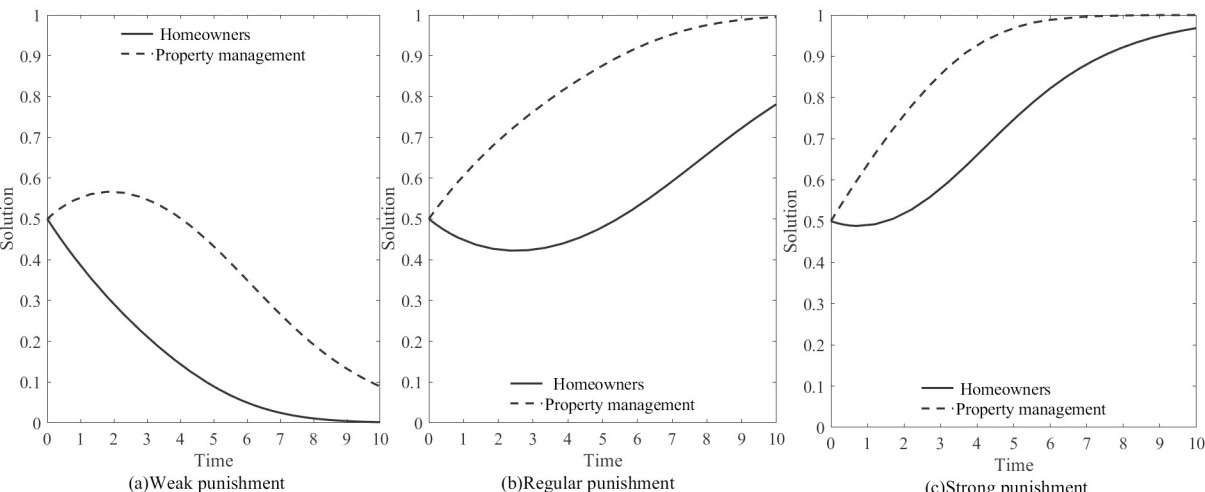

**Fig 6. Evolution of the game under Government Punishment.**

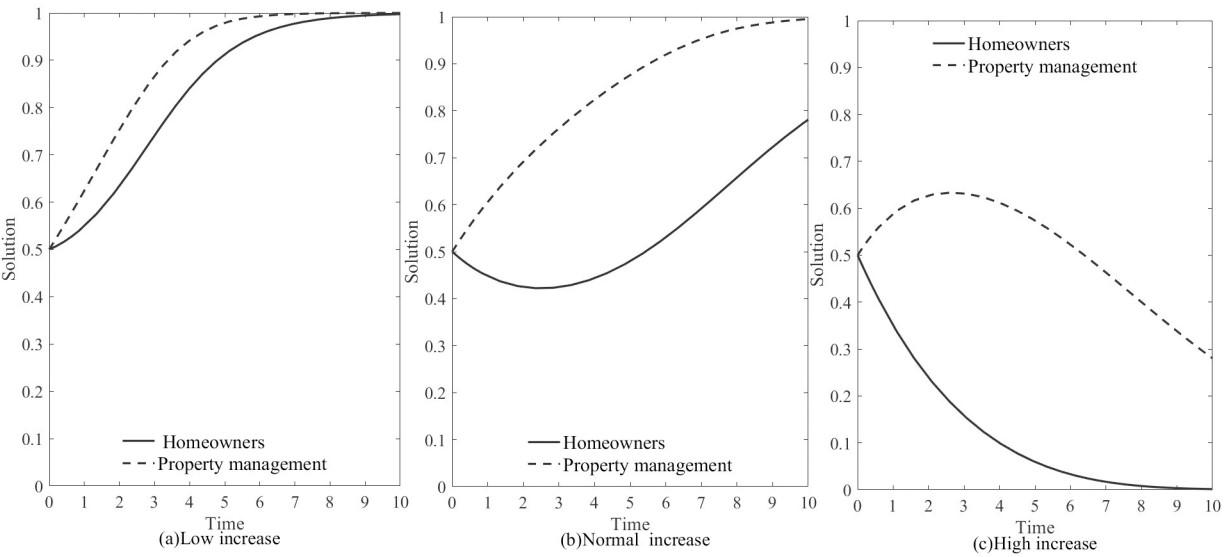

**Fig 7. Evolution of the game under property fee increase.**

increase", "Normal increase", and "High increase" respectively.in property fees.set the following conditions. The impact of property fee increase on the game is defined as follows:

$$\Delta C = 1.6, Z = 0.25, B = 12, H_1 = 3.6, H_2 = 3, D = 1.2, A = 1.5, (0.5, 0.5);$$

$$\Delta C = 2, Z = 0.25, B = 12, H_1 = 4.5, H_2 = 3.7, D = 1.2, A = 1.5, (0.5, 0.5);$$

$$\Delta C = 2.4, Z = 0.25, B = 12, H_1 = 5.1, H_2 = 4.1, D = 1.2, A = 1.5, (0.5, 0.5)$$

Fig 7 show the evolution of the game under property fee increase. When the property proposes a price increase, the stable evolutionary strategy tends towards (1, 1). After a period of hesitation, both parties ultimately choose to "Cooperate". When the property fee increase is high, the stable evolutionary strategy tends towards (0, 0), exceeding the homeowners' acceptance range. Regardless of how much the property service quality improves, the homeowners' decision is to "Betray", and both parties will eventually choose "Betrayal". When the property fee increase is low, the stable evolutionary strategy tends towards (1, 1), as the homeowners can accept the increase within this range. Therefore, there is no hesitation in the decision-making process, and both parties choose to "Cooperate". Thus, appropriately reducing the property fee increase can eliminate the homeowners' hesitation in decision-making, enabling both parties to reach "Cooperation" more quickly.

## VI. Conclusion and recommendations

The information asymmetry between homeowners and property management leads to difficulties in adjusting property fees. Traditional regulatory measures are insufficient to solve this problem, while the existence of "Smart regulation" helps to address these limitations. This paper explores the impact of "Smart regulation" technology on property fee adjustments. Firstly, a cooperative game model between homeowners and property management under traditional regulation is constructed, and the reasons for the challenges in property fee adjustments are analyzed. Then, the concept of "Smart regulation" is introduced, and an evolutionary game model driven by smart regulation technology is built. The model is solved,

and stability analysis is conducted, comparing the results with those of traditional games. Finally, numerical simulations using Matlab are conducted to validate the accuracy of the model and further explore the changes in the game system after the establishment of a smart regulation platform.

The following specific conclusions are drawn in this paper:

1. Under traditional regulation, in the cooperative game between both parties, As both parties are rational actors seeking maximum benefits, their interests conflict, leading to the "Prisoner's dilemma" in which property service companies and homeowners easily fall into a pricing predicament. The current traditional regulatory methods struggle to accurately assess information such as the service quality and cost expenditure of property service companies. The lack of understanding between both parties means that their contradictions cannot be effectively resolved, ultimately resulting in a pricing predicament for property fees. Traditional regulatory methods are ineffective in addressing the issue of information asymmetry, leading to challenges in property fee adjustments. Therefore, it is necessary to introduce new theories to improve regulatory mechanisms and resolve the pricing predicament between property service companies and homeowners.

2. In the game simulation study on property fee adjustments, the results indicate that the establishment of an intelligent oversight platform has a positive effect on promoting cooperative decision-making between both parties. Furthermore, the study reveals that there is a positive correlation between the government's level of regulation and punishment and the decision-making of both parties after the platform is established, indicating that stronger regulation and punishment can encourage a greater inclination towards "Cooperation". These research findings suggest that introducing an intelligent oversight platform and strengthening government regulation and punishment can effectively enhance the willingness and behavior of cooperation in property cost adjustments, thereby facilitating a smooth adjustment process.

3. In the simulation game, it was also found that the growth rate of property service fees also has an impact on the decision-making of both parties. When the growth rate of property service fees is too high, it is difficult for the group of homeowners to accept. Therefore, it is crucial to balance the growth of property service fees in the property cost adjustment process. A reasonable level of property service fee growth can meet the operational costs of the property management side and at the same time safeguard the interests of homeowners within a reasonable range. Through fair adjustments of property service fees, it is possible to promote cooperation and coordination between homeowners and property management, thereby reducing potential conflicts and disputes. Therefore, when formulating property cost adjustment policies, it is necessary to consider the interests and affordability of all parties and establish reasonable restrictions and adjustment mechanisms for the growth of property service fees to achieve cooperation and win-win outcomes for both sides.

4. The traditional regulatory approach has led to information asymmetry, which is also one of the reasons for the difficulty in adjusting property expenses. Through game analysis, we conclude that intelligent regulatory platforms can solve the problem of information asymmetry by improving transparency, thereby promoting the success of property expense adjustments. Information asymmetry is a common challenge in many industries, and the successful application of intelligent regulatory platforms in one industry can also provide similar solutions for other industries. For example, by real-time monitoring of household electricity and water consumption and sharing the data with homeowners and property management companies, intelligent regulatory platforms can avoid information asymmetry

and ensure the fairness and reasonableness of expense adjustments. In summary, the application of intelligent regulatory platforms can solve the problem of information asymmetry, promote cooperation and trust among all parties, and provide important reference for solving similar problems in other industries.

This study introduces the "intelligent supervision" technology to address the issue of property fee adjustment, which has important theoretical, practical, and promotional value. Theoretically, this study reveals the limitations of traditional regulatory methods and proposes a new solution, enriching the research content in the field of property management. From a practical perspective, intelligent supervision technology can improve the fairness and efficiency of property fee adjustment, helping to enhance the quality of property services and owner satisfaction. From a promotional perspective, intelligent supervision technology can also be applied to other industries to address the problems of information asymmetry and regulatory difficulties. In conclusion, this study has important value in terms of theory, practice, and promotion.

## Supporting information

**S1 File.**
(ZIP)

## Author Contributions

**Conceptualization:** Dekun Dong.

**Writing – original draft:** Shuang Li.

**Writing – review & editing:** Dekun Dong.

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
