## [Decision Letter · Decision Letter 0]

6 Oct 2023

PONE-D-23-27714How a smart regulatory platform can improve the property fee adjustment dilemmaPLOS ONE

Dear Dr. Li,

Thank you for submitting your manuscript to PLOS ONE. After careful consideration, we feel that it has merit but does not fully meet PLOS ONE’s publication criteria as it currently stands. Therefore, we invite you to submit a revised version of the manuscript that addresses the points raised during the review process.

I recommend that it should be revised taking into account the changes requested by the reviewers. Since the requested changes include valuable and constructive reviews, I would like to give you a chance to revise your manuscript. The revised manuscript will undergo the next round of review by same reviewers.

We look forward to receiving your revised manuscript.

Kind regards,

Baogui Xin, Ph.D.

Academic Editor

PLOS ONE

Journal Requirements:

NO authors have competing interests.

Additional Editor Comments:

1.Overall Comments:

This paper provides an interesting analysis of the property fee adjustment dilemma using evolutionary game theory models. The topic is novel and highly relevant given the practical issues with property service pricing. The use of game theory and simulation methods is technically sound. The manuscript would benefit from some additional background context and discussion of real-world implementation.

2.Advantages:

- Evolutionary game theory is an appropriate method for modeling the dynamic interactions between homeowners, property companies, and government over time. The models are logically constructed.

- The literature review covers relevant research areas including game theory applications and property fee studies. This provides helpful context.

- Assumptions made in building the models are reasonably stated. Key variables are well defined mathematically.

- Stability analysis of the evolutionary game model under smart regulation provides useful equilibrium insights.

- The Matlab simulations demonstrate model applicability and explore the effects of parameters like fee increases. This strengthens the practical relevance of the analysis.

3.Disadvantages:

- More background details would help frame the problem context - information on study location, property management practices, and fee setting policies.

- Practical implementation considerations around adoption, costs, and challenges of the proposed smart platform are not extensively discussed.

- The models assume rational behavior, but irrational biases are common in pricing dilemmas. This aspect could be incorporated.

- Broadening the implications and applications of the results beyond property fee adjustments would improve scope.

4.Suggested Modifications:

- Provide more background details on the specific property management context related to the study area.

- Expand the literature review to include behavioral economics concepts like irrational biases relevant to pricing games.

- Discuss practical implementation, adoption strategies, costs and challenges associated with the proposed smart regulatory platform.

- The Discussion needs to be a coherent and cohesive set of arguments that take us beyond this study in particular and help us see the relevance of what the authors have proposed. The authors need to contextualize the findings in the literature and need to be explicit about the added value of your study towards that literature. Also, other studies should be cited to increase the theoretical background of each of the methods used. Findings should be contextualized in the literature and should be explicit about the added value of the study towards the literature. The contribution and implications of the article are yet to be specified. Please refer to the style, DOI: 10.1016/j.renene.2023.118960 or 10.1007/s10479-023-05334-1

- In the conclusion, broaden the implications of results to other pricing dilemmas like utility bills, tuition fees etc. beyond just property management.

- Carefully proofread the manuscript to fix minor grammatical errors.

Reviewers' comments:

Reviewer's Responses to Questions

**Comments to the Author**

1. Is the manuscript technically sound, and do the data support the conclusions?

Reviewer #1: Yes

Reviewer #2: Yes

Reviewer #3: Yes

2. Has the statistical analysis been performed appropriately and rigorously? 

Reviewer #1: Yes

Reviewer #2: Yes

Reviewer #3: Yes

3. Have the authors made all data underlying the findings in their manuscript fully available?

Reviewer #1: Yes

Reviewer #2: Yes

Reviewer #3: Yes

4. Is the manuscript presented in an intelligible fashion and written in standard English?

Reviewer #1: Yes

Reviewer #2: Yes

Reviewer #3: Yes

5. Review Comments to the Author

Reviewer #1: 1.Overall Comments:

This paper provides an interesting analysis of the property fee adjustment dilemma using evolutionary game theory models. The topic is novel and highly relevant given the practical issues with property service pricing. The use of game theory and simulation methods is technically sound. The manuscript would benefit from some additional background context and discussion of real-world implementation.

2.Advantages:

- Evolutionary game theory is an appropriate method for modeling the dynamic interactions between homeowners, property companies, and government over time. The models are logically constructed.

- The literature review covers relevant research areas including game theory applications and property fee studies. This provides helpful context.

- Assumptions made in building the models are reasonably stated. Key variables are well defined mathematically.

- Stability analysis of the evolutionary game model under smart regulation provides useful equilibrium insights.

- The Matlab simulations demonstrate model applicability and explore the effects of parameters like fee increases. This strengthens the practical relevance of the analysis.

3.Disadvantages:

- More background details would help frame the problem context - information on study location, property management practices, and fee setting policies.

- Practical implementation considerations around adoption, costs, and challenges of the proposed smart platform are not extensively discussed.

- The models assume rational behavior, but irrational biases are common in pricing dilemmas. This aspect could be incorporated.

- Broadening the implications and applications of the results beyond property fee adjustments would improve scope.

4.Suggested Modifications:

- Provide more background details on the specific property management context related to the study area.

- Expand the literature review to include behavioral economics concepts like irrational biases relevant to pricing games.

- Discuss practical implementation, adoption strategies, costs and challenges associated with the proposed smart regulatory platform.

- The Discussion needs to be a coherent and cohesive set of arguments that take us beyond this study in particular and help us see the relevance of what the authors have proposed. The authors need to contextualize the findings in the literature and need to be explicit about the added value of your study towards that literature. Also, other studies should be cited to increase the theoretical background of each of the methods used. Findings should be contextualized in the literature and should be explicit about the added value of the study towards the literature. The contribution and implications of the article are yet to be specified. Please refer to the style, DOI: 10.1016/j.renene.2023.118960 or 10.1007/s10479-023-05334-1

- In the conclusion, broaden the implications of results to other pricing dilemmas like utility bills, tuition fees etc. beyond just property management.

- Carefully proofread the manuscript to fix minor grammatical errors.

Reviewer #2: 1. It is recommended to change the subtitle to "MatLAB-based Evolutionary Game Analysis" or delete the subtitle.

2. "with the increase in the use time of buildings, the cost of daily maintenance and repair will also increase to some extent" Should be amended to" As buildings last longer, the cost of routine maintenance increases”.

3. Reference 11 was published in 2013, “For example, in 2021, a district in Qingdao received nearly 7,000 complaints. Among the 237 communities under the jurisdiction of this district, 113 communities had property fees lower than 0.6 yuan, accounting for 47%." There is a problem with the annotation of the statement literature, please re-annotate or modify.

4. "Keyword" should be "Keywords".

Reviewer #3: General Comments:

This manuscript explores the problem that property companies face revenue difficulties due to the inability to adjust property service fees synchronously. And then, analyzes the information asymmetry dilemma in the process of property fee adjustment, constructs an evolutionary game model of government, property, and owners, and uses Matlab for simulation. The logical structure of this manuscript is reasonable, and the mastery of various concepts and tools is relatively proficient. The research also has certain application value. The content of the article needs to be carefully revised to ensure that the article is more reasonable.

1. In keywords, the capitalization of the first letter needs to be consistent. Please determine how to write "Evolutionary Game".

2. In the literature review section, please confirm whether the author of the cited literature needs to be given his or her full name, and then modify it.

3. There are many formulas in the article, which can well explain the content of the article. But some formulas contain editing errors. Please check each formula to ensure its correctness. And make sure the format of the formula is consistent and try to make it as simple and beautiful as possible.

4. The labeling of the chart is a bit confusing. First of all, the first picture is named "Image 1", which does not correspond to the "figure" below. Moreover, "Figure 2" appears twice in the manuscript. Please re-layout to ensure that the order of the charts is correct and reasonable.

5. The content of the chart also needs to be modified to ensure that its introduction is beautiful. Taking "Figure 1" as an example, it is best to keep the arrow sizes the same. The meanings of the abscissa and ordinate in "Figure 2" need to be clearly marked in the figure. Other charts should also be modified appropriately to ensure that readers can understand them.

6. The format of references is not unified. For example, the citation format of articles 4, 5, and 6 is obviously different from other documents. Even the author's name is not displayed in the 13th article. I don't know if it's my fault. Please follow the prescribed document citation format and carefully modify the format of the reference to ensure the uniformity of the citation format.

7. There are also some writing errors in the article, such as the capitalization of the first letter, the missing left bracket, etc. Please revise the article content carefully to avoid simple errors.

6. PLOS authors have the option to publish the peer review history of their article (what does this mean?). If published, this will include your full peer review and any attached files.

Reviewer #1: No

Reviewer #2: No

Reviewer #3: No

---

## [Author Response · Author response to Decision Letter 0]

31 Oct 2023

Reviewer #1:

1.Provide more background details on the specific property management context related to the study area.

Thank you for your suggestions. We have made the corresponding modifications in the background section, added specific details:and incorporated relevant references to enhance the description and elucidation of the property management background in the research area.

Specifically:Property management part of this paper supplements the research of scholars on the status quo of the property survival environment and adds examples to support it, and the smart regulation part of this paper supplements the suggestions put forward by other scholars for the improvement of government regulation.

2.Expand the literature review to include behavioral economics concepts like irrational biases relevant to pricing games.

Thank you for your suggestions. In the literature review section, we have added a discussion on game theory and specifically examined the aspect of irrationality in the game process to broaden the scope of the literature review. 

3.Discuss practical implementation, adoption strategies, costs and challenges associated with the proposed smart regulatory platform.

Thank you for your suggestions.In the theoretical analysis section of Chapter 4, we have supplemented the relevant details of "smart supervision".

The additional paragraph in the theoretical analysis section is as follows:The government plays a crucial role in community governance and should be responsible for establishing a publicly available information disclosure platform to ensure transparency in property service quality and costs. At the same time, property companies should regularly report their detailed income breakdowns and the deployment status of community staff to this platform. The government can assess the quality of property services based on this data and provide reasonable market-adjusted prices. Homeowners can use the platform to report property issues to the government, which can conveniently collect and organize these complaints, conduct regular inspections based on homeowner feedback, and impose penalties on underperforming property companies.The specific operational concept of the platform in this paper is shown in Fig 1.

4. The Discussion needs to be a coherent and cohesive set of arguments that take us beyond this study in particular and help us see the relevance of what the authors have proposed. The authors need to contextualize the findings in the literature and need to be explicit about the added value of your study towards that literature. Also, other studies should be cited to increase the theoretical background of each of the methods used. Findings should be contextualized in the literature and should be explicit about the added value of the study towards the literature. The contribution and implications of the article are yet to be specified. Please refer to the style, DOI: 10.1016/j.renene.2023.118960 or 10.1007/s10479-023-05334-1

Thank you for your suggestions. In the conclusion, we have combined our research to explain the value of this article in terms of theoretical contribution, practical significance, and research insights.

The Conclusion section has been revised as follows:This study introduces the "intelligent supervision" technology to address the issue of property fee adjustment, which has important theoretical, practical, and promotional value. Theoretically, this study reveals the limitations of traditional regulatory methods and proposes a new solution, enriching the research content in the field of property management. From a practical perspective, intelligent supervision technology can improve the fairness and efficiency of property fee adjustment, helping to enhance the quality of property services and owner satisfaction. From a promotional perspective, intelligent supervision technology can also be applied to other industries to address the problems of information asymmetry and regulatory difficulties. In conclusion, this study has important value in terms of theory, practice, and promotion.

5.In the conclusion, broaden the implications of results to other pricing dilemmas like utility bills, tuition fees etc. beyond just property management.

Thank you for your suggestions.In the conclusion, we expanded the scope of the impact of our research findings.

The additional paragraph in the Conclusion section is as follows:

The traditional regulatory approach has led to information asymmetry, which is also one of the reasons for the difficulty in adjusting property expenses. Through game analysis, we conclude that intelligent regulatory platforms can solve the problem of information asymmetry by improving transparency, thereby promoting the success of property expense adjustments. Information asymmetry is a common challenge in many industries, and the successful application of intelligent regulatory platforms in one industry can also provide similar solutions for other industries. For example, by real-time monitoring of household electricity and water consumption and sharing the data with homeowners and property management companies, intelligent regulatory platforms can avoid information asymmetry and ensure the fairness and reasonableness of expense adjustments. In summary, the application of intelligent regulatory platforms can solve the problem of information asymmetry, promote cooperation and trust among all parties, and provide important reference for solving similar problems in other industries.

6.Carefully proofread the manuscript to fix minor grammatical errors.

Thank you for your suggestions，We carefully proofread the manuscript and corrected any grammar errors.

Reviewer #2: 

1.It is recommended to change the subtitle to "MatLAB-based Evolutionary Game Analysis" or delete the subtitle.

Thank you for your suggestion .We have deleted the subtitle.

2."with the increase in the use time of buildings, the cost of daily maintenance and repair will also increase to some extent" Should be amended to" As buildings last longer, the cost of routine maintenance increases”.

Thank you for your suggestion. "with the increase in the use time of buildings, the cost of daily maintenance and repair will also increase to some extent" has been modified to " As buildings last longer, the cost of routine maintenance increases”.

3.Reference 11 was published in 2013, “For example, in 2021, a district in Qingdao received nearly 7,000 complaints. Among the 237 communities under the jurisdiction of this district, 113 communities had property fees lower than 0.6 yuan, accounting for 47%." There is a problem with the annotation of the statement literature, please re-annotate or modify.

Thank you for your suggestions.The reference literature is a summary of the phenomenon of low property fees, and this issue was raised in 2013. In order to make the argument more relevant to reality, this article uses a community case study in 2021 as a real-life example, and there is no time conflict between the two. The narrative in the article indeed had problems, causing confusion in logic, and has now been modified.

The article is now described as follows:In addition, Research has found that in situations where property service fees are low, property companies often deliberately and selectively ignore negative feedback from homeowners. This practice exacerbates conflicts between homeowners and property companies. For example, in 2021, a district in Qingdao received nearly 7,000 complaints. Among the 237 communities under the jurisdiction of this district, 113 communities had property fees lower than 0.6 yuan, accounting for 47%. Therefore, long-term low property fees are not conducive to community governance, and property fees need to be adjusted. 

4."Keyword" should be "Keywords".

Thank you for your suggestions.We have made the necessary revisions as requested."Keyword"has been modified to "Keywords".

Reviewer #3:

1.In keywords, the capitalization of the first letter needs to be consistent. Please determine how to write "Evolutionary Game".

Thank you for your suggestions.We have fixed the formatting issue with the keywords."Evolutionary Game"has been modified to"Evolutionary game"

2.In the literature review section, please confirm whether the author of the cited literature needs to be given his or her full name, and then modify it.

Thank you for your suggestions.We have standardized the format of references.

3.There are many formulas in the article, which can well explain the content of the article. But some formulas contain editing errors. Please check each formula to ensure its correctness. And make sure the format of the formula is consistent and try to make it as simple and beautiful as possible.

Thank you for your suggestions.We have modified and simplified the equations in the article according to the requirements of the journal.

4.The labeling of the chart is a bit confusing. First of all, the first picture is named "Image 1", which does not correspond to the "figure" below. Moreover, "Figure 2" appears twice in the manuscript. Please re-layout to ensure that the order of the charts is correct and reasonable.

Thank you for your suggestions.We have revised and aligned the tables and figures in the article to the requirements of the journal.According to the journal's requirements, the labels for the figures have been standardized. The label for the images is "Fig" and the label for the tables is "Table".

5.The content of the chart also needs to be modified to ensure that its introduction is beautiful. Taking "Figure 1" as an example, it is best to keep the arrow sizes the same. The meanings of the abscissa and ordinate in "Figure 2" need to be clearly marked in the figure. Other charts should also be modified appropriately to ensure that readers can understand them.

Thank you for your suggestions.The arrows in Fig 1 have been standardized. As for Fig 2 and other graphs, the X and Y coordinates have been labeled accordingly.

6.The format of references is not unified. For example, the citation format of articles 4, 5, and 6 is obviously different from other documents. Even the author's name is not displayed in the 13th article. I don't know if it's my fault. Please follow the prescribed document citation format and carefully modify the format of the reference to ensure the uniformity of the citation format.

Thank you for your suggestions.We have standardized the format of references.

7.There are also some writing errors in the article, such as the capitalization of the first letter, the missing left bracket, etc. Please revise the article content carefully to avoid simple errors.

Thank you for your suggestions.We have already made corrections to the spelling errors found in the article.

---

## [Decision Letter · Decision Letter 1]

13 Nov 2023

How a “Smart regulatory" platform can improve the property fee adjustment dilemma

PONE-D-23-27714R1

Dear Dr. Li,

We’re pleased to inform you that your manuscript has been judged scientifically suitable for publication and will be formally accepted for publication once it meets all outstanding technical requirements.

Kind regards,

Baogui Xin, Ph.D.

Academic Editor

PLOS ONE

Additional Editor Comments (optional):

Reviewers' comments:

Reviewer's Responses to Questions

**Comments to the Author**

1. If the authors have adequately addressed your comments raised in a previous round of review and you feel that this manuscript is now acceptable for publication, you may indicate that here to bypass the “Comments to the Author” section, enter your conflict of interest statement in the “Confidential to Editor” section, and submit your "Accept" recommendation.

Reviewer #1: (No Response)

2. Is the manuscript technically sound, and do the data support the conclusions?

Reviewer #1: (No Response)

3. Has the statistical analysis been performed appropriately and rigorously? 

Reviewer #1: (No Response)

4. Have the authors made all data underlying the findings in their manuscript fully available?

Reviewer #1: (No Response)

5. Is the manuscript presented in an intelligible fashion and written in standard English?

Reviewer #1: (No Response)

6. Review Comments to the Author

Reviewer #1: (No Response)

7. PLOS authors have the option to publish the peer review history of their article (what does this mean?). If published, this will include your full peer review and any attached files.

Reviewer #1: No

---

## [Editor Report · Acceptance letter]

21 Nov 2023

PONE-D-23-27714R1 

How a “Smart regulatory" platform can improve the property fee adjustment dilemma 

Dear Dr. Li:

I'm pleased to inform you that your manuscript has been deemed suitable for publication in PLOS ONE. Congratulations! Your manuscript is now with our production department. 

Kind regards, 

on behalf of

Professor Baogui Xin 

Academic Editor

PLOS ONE